# Dynamic Gesture Recognition Based on FMCW Millimeter Wave Radar: Review of Methodologies and Results

**DOI:** 10.3390/s23177478

**Published:** 2023-08-28

**Authors:** Gaopeng Tang, Tongning Wu, Congsheng Li

**Affiliations:** China Academy of Information and Communications Technology, Beijing 100191, China; tgp1072564908@163.com (G.T.); wutongning@caict.ac.cn (T.W.)

**Keywords:** gesture recognition, FMCW millimeter-wave radar, feature extraction, classification, generalization

## Abstract

As a convenient and natural way of human-computer interaction, gesture recognition technology has broad research and application prospects in many fields, such as intelligent perception and virtual reality. This paper summarized the relevant literature on gesture recognition using Frequency Modulated Continuous Wave (FMCW) millimeter-wave radar from January 2015 to June 2023. In the manuscript, the widely used methods involved in data acquisition, data processing, and classification in gesture recognition were systematically investigated. This paper counts the information related to FMCW millimeter wave radar, gestures, data sets, and the methods and results in feature extraction and classification. Based on the statistical data, we provided analysis and recommendations for other researchers. Key issues in the studies of current gesture recognition, including feature fusion, classification algorithms, and generalization, were summarized and discussed. Finally, this paper discussed the incapability of the current gesture recognition technologies in complex practical scenes and their real-time performance for future development.

## 1. Introduction

In recent years, with the continuous development of intelligent perception and human-computer interaction technologies, gesture recognition has received more attention and has been used as a convenient approach to human-computer interaction [1] in many fields, including smart homes [2], smart vehicles [3], sign language communication [4], electronic device control [5], games, and virtual reality [6]. In its early stages, gesture recognition usually relies on wearable sensors [7], such as data gloves [8], surface electromyography sensors [9], accelerometer and gyroscope sensors [10], and wearable sensors based on photoplethysmography [11], which also have good recognition performance. These sensors are able to obtain a wealth of information about the operator’s hand movements. However, gesture recognition technology based on wearable sensors is cumbersome and expensive, which often leads to inconvenience for users and has not been widely used in daily life [12]. Therefore, gesture recognition based on contactless sensing has attracted more attention, such as computer vision methods using RGB and depth images [13] and radio frequency identification based on WiFi and radar signals [14]. A computer vision-based gesture recognition method collects images of dynamic gestures and recognizes gestures based on features such as appearance, contour, or skeleton of the gesture, which has high recognition accuracy [15]. With the advancement of depth sensing technology, gesture recognition based on depth cameras such as Kinect [16], RealSense [17], and Leap Motion [18] has received widespread attention, which can achieve more accurate and robust recognition than traditional cameras and can be applied to complex 3D gesture recognition. Depth cameras can provide real-time tracking of gestures and movements, allowing for immediate responses and interactions. However, this method is highly dependent on the brightness of environmental conditions [19]. To note, it requires much computational resources in dynamic gesture recognition [20] and brings potential leakage of privacy. The WiFi-based method uses Channel State Information (CSI) and Received Signal Strength Indicator (RSSI) as features for gesture recognition, but this method is susceptible to interference and makes it difficult to recognize complex gestures [21]. LiDAR [22] is a sensor that utilizes infrared light to determine the distance between the sensor and an object by projecting a pulse of laser light, which is highly accurate in ranging and has a higher level of safety compared to cameras. In addition, LiDAR is not reliant on ambient light and can operate effectively in low light or complete darkness. In gesture recognition, LiDAR can be used to capture 3D point clouds of hand movements and recognize different gestures, enabling touchless interactions with devices or virtual environments. However, LiDAR is not sensitive to complex gesture changes and is susceptible to occlusions. As a consequence, millimeter-wave (mmW) radar-based sensing became an option. Millimeter wave radar combines the advantages of microwave radar and LiDAR in terms of privacy protection, light robustness, small size, low cost, and convenience during gesture recognition [23]. Further, mmW radar has a variety of waveforms, such as Continuous Wave (CW), Frequency-shift keying (FSK), and Frequency-Modulated Continuous Wave (FMCW). FMCW mmW radar offers higher accuracy, robustness, and efficiency compared to other waveforms, which have been widely used in gesture recognition [24].

At present, the studies relating to FMCW mmW radar-based gesture recognition have achieved certain milestones. For example, in 2015, Google’s Soli project implemented proximity micro-motion gesture recognition by end-to-end convolutional recurrent neural networks based on distance Doppler features using a FMCW mmW radar chip at 60 GHz [25], and this study demonstrated the capability of FMCW mmW radar for this application of gesture recognition. Although there are a number of studies on gesture recognition based on FMCW mmW radar, systematic analysis of the current method is scarce.

This paper summarized the main methods and challenges involved in recent research on FMCW mmW radar-based gesture recognition and discussed the key issues as well as the development of the technology.

The remainder of this paper is organized as follows: Section 2 introduces the searching strategy. Section 3 introduces information about FMCW mmW radar and gestures. Section 4 summarizes the methods and results in data processing and classification, analyzes the statistics, gives recommendations, and discusses the key issues involved. Section 5 discusses the challenges of gesture recognition. Section 6 concludes this paper. 

## 2. Searching Strategy

In this review article, we searched several databases for references and covered the following databases:(1)Web of Science;(2)IEEE Explore Digital Library(3)Association for Computing Machinery;(4)Springer Link;(5)Google Scholar.

The searching keyword was a combination of “Gesture recognition”, “Radar”, “Millimeter-wave”, and “FMCW”.

To screen our initial searches, we applied the following inclusion criteria:(1)Publication date: between January 2015 and January 2023.(2)Searching domain: science, technology, or computer science.(3)Publication types: journals, proceedings, and conferences.(4)Language: English.

We applied the following exclusion criteria:(1)Studies that do not include FMCW mmW radar-based gesture recognition.(2)Studies that do not provide details of experiments or experimental designs.(3)Studies that replicate with others.(4)Studies for which the full text of the paper is not available.

In addition, we classified the studies according to their publication date, innovation, accuracy, features, and algorithms in order to compare and summarize the problems, methods, and results involved.

## 3. FMCW mmW Radar and Gestures

In general, FMCW MMW radar-based gesture recognition is divided into three main steps: data acquisition, data processing, and classification. At the beginning of any study, researchers need to select the appropriate FMCW mmW radar and gestures as tools and objects for data acquisition. In this section, we introduced FMCW mmW radar and gestures in gesture recognition.

### 3.1. FMCW mmW Radar

In gesture recognition, the FMCW mmW radar transmits FM continuous waves, whose frequency increases over time, to the hand. The received and transmitted signals are filtered through a mixer and low-pass filter. After ADC sampling, an intermediate frequency (IF) signal is generated. The IF signal allows information such as the distance, speed, and even angle of the gesture to be calculated and the corresponding feature maps to be obtained. These features reflect the motion of the gesture through changes in distance, Doppler frequency, and angle rather than 3D shape, which is one of the differences between dynamic and static gesture recognition. The whole process of data acquisition is shown in Figure 1.

The range measurement in FMCW radar is based on the time delay between the transmitted and received signals. The range of the target (R) can be calculated using the formula:$$R=\frac{c\cdot \tau }{2}=f_{IF}\frac{cT}{2B}$$
where c is the speed of light, τ is the time delay between the transmitted and received signals, $f_{IF}$ is the frequency of the IF signal, and B and T are the bandwidth and the sweep period of the chirp signal. This process can be achieved by FFT for the IF signal.

The range resolution of the FMCW radar can be derived as follows:$$R_{res}=\frac{c}{2B}$$

Therefore, the range resolution of the radar can be improved by increasing the bandwidth.

The velocity measurement in FMCW radar is based on the Doppler effect, which can be obtained by using phase difference. The velocity of the target (v) can be calculated using the formula:$$v=\frac{\lambda \Delta \phi }{4\pi T}$$
where λ is the wavelength of the chirp signal and Δϕ is the phase difference. This process can be achieved by using 2D-FFT for the IF signal.

The velocity resolution of the FMCW radar can be derived as follows:$$v_{res}=\frac{\lambda }{2NT}$$
where N is the number of chirps in a frame.

The angle measurement in FMCW radar is typically achieved using an antenna array with multiple elements. The radar can estimate the angle of the target based on the phase differences between the signals at different antennas. The angle of the target (θ) can be calculated using the formula:$$\theta =\sin^{-1}(\frac{\lambda \Delta \phi }{2\pi l})$$
where l is the distance between adjacent antenna elements.

The angular resolution of the FMCW radar can be derived as follows:$$\theta_{res}=\frac{\lambda }{N_{R}l\cos(\theta )}$$
where $N_{R}$ is the number of RX antennas, and the angular resolution can be improved by increasing the number of antennas.

In order to understand the use of FMCW mmW radars, we summarized the information on FMCW mmW radars used in numerous studies in recent years (Table 1). It can be found that the FMCW mmW radars used in all the studies we investigated were concentrated in several types of radars, as shown in Table 1. 24 GHz, 60 GHz, and 77 GHz are the main frequency bands used by millimeter-wave radars at present, and the radars shown in Table 1 are also distributed in these three frequency bands. It is worth noting that radars covering 76–81 GHz and 60–64 GHz are favored for most studies in gesture recognition, while radars operating at 24 GHz are rarely used. We attribute this situation to the performance of the radars in terms of range resolution and velocity resolution. According to the resolution equations for distance and velocity, we know that the radars in the frequency bands 76–81 GHz and 60–64 GHz (Table 1) tend to provide higher resolution in gesture recognition due to their high frequency and wide bandwidth, which in turn improves the accuracy of the recognition results. In addition, gesture recognition usually requires angle information in the process of target motion, which requires the radar used for data acquisition to have multiple receiving antennas, as can also be proven by Table 1. It is worth noting that Wu et al. [26] tested the effect of using different numbers of receiving antennas (1, 2, and 4) for gesture recognition and found that more receiving antennas used to collect gesture data can often obtain higher recognition accuracy. This result is also consistent with the method we mentioned to improve the angular resolution. 

In addition to the information in Table 1, we also counted the power, gain, maximum, and minimum detection range of these radars according to the datasheet. We found that the power and gain values of these radars are designed in a small range; for example, the TX power is all in the range of 10–15 dBm, with very little difference from each other. Similarly, except for the BGT24MTR12, which has a minimum detection range of 0.5 m, all the other radars achieve a minimum detection range of centimeters and a maximum detection range of more than 10 m. In fact, since gesture recognition based on FMCW millimeter-wave radar is generally in the detection scene at close range, it is not sensitive to the above parameters compared with frequency, bandwidth, and the number of antennas.

### 3.2. Gestures

In gesture recognition, the gestures in the dataset need to be predefined. In fact, the complexity of the gesture is an important factor affecting recognition accuracy. The choice of gestures is mostly determined by the experimental context and the conditions of the study. However, the complexity of gestures varies, which affects the comparison and evaluation of the results of different experiments. Therefore, a systematic summary and classification of dynamic gestures is necessary. 

The gestures chosen for the experiments should be generic and distinctive, taking into account the differences in gestures due to individual habits. We counted the gestures to be tested that were of concern in previous studies, as shown in Figure 2, which appeared in more than two papers. These gestures are considered to be divided into macro gestures and micro gestures [39]. Macro gestures usually take the palm movement as the main recognition subject, excluding finger movements, and are defined according to the changing process of the spatial position of the palm. In the statistical process, we attempt to further classify macro gestures into those with single-direction (Figure 2a) and those with multi-direction (Figure 2b). This is due to the fact that macro gestures with multi-directions were found to be more likely to be misclassified in several studies because a certain part of the movement is more prominent and more similar to single-directional gestures [40,41]. This classification also provides a better statistical measure of the complexity of gestures. Micro-gestures usually take finger movements as the main recognition subject, and micro-gestures often do not include the spatial position changes of the palm. Since the radar reflection area and motion amplitude of fingers are smaller than those of palms, the features of microscopic gestures are weaker and more susceptible to interference from other reflected signals, resulting in misjudgment. Therefore, the recognition of micro gestures has higher requirements for feature extraction, clutter suppression, and classification algorithms.

In addition, dynamic gestures are divided into isolated gestures and continuous gestures. This is a definition of gesture types based on coherence. Currently, most of the research on gesture recognition based on radar sensors uses isolated gestures [42]. This is due to the fact that isolated gestures have significant action boundaries and are easy to detect and recognize. In contrast to isolated gestures, continuous gestures can improve the speed and efficiency of gesture recognition. However, the accurate segmentation of continuous gestures is a challenge for recognizing continuous gestures [43], which largely increases the difficulty of accurate gesture recognition. It is necessary to determine the beginning and end of a gesture according to the features of gestures so as to realize the segmentation of continuous gestures. Zhou et al. [44] obtained the total time of a single hand gesture and realized the detection of continuous gestures. Ren et al. [45] obtained the amplitude by normalizing the hand gesture target and setting a threshold to effectively segment the continuous gestures.

## 4. Methodologies

After data acquisition, the raw data needs to be processed to extract the gesture features and build the dataset for training and testing the classification model (Figure 3). Finally, the features are classified by a classification algorithm to obtain the results of gesture recognition. Feature extraction and classification algorithms are the most important parts of gesture recognition and have been the focus of researchers. In this section, we summarized the methods in gesture recognition and discussed and made suggestions on the problems of feature fusion, clutter suppression, and generalization based on the statistical results.

### 4.1. Pre-Processing

The raw gesture data contains clutter that can seriously interfere with gesture recognition. It is necessary to pre-process the acquired data to remove interference while retaining the main gesture data. In this section, we summarize the conventional pre-processing methods for radar data. Since Range Doppler Map (RDM) is the classical way to describe single frames of signal data from FMCW mmW radar, which can significantly show the clutter distribution, we introduced the method of acquiring RDM and summarized the main methods for removing clutter based on previous studies.

In general, the gesture signal needs to be processed into a data matrix, the rows of which represent the sampled values of the chirp signal in the fast-time and slow-time domains, respectively. The popular pre-processing method for the chirp signal is the Fourier transform, such as the fast Fourier transform (FFT) [46] and the short-time Fourier transform (STFT) [47]. For each data matrix, an FFT is performed in the fast-time domain (the IF signal sampling direction) to obtain a two-dimensional range map. By performing FFT on the 2D range map in the slow-time domain (the chirp index direction), the RDM can be obtained [48]. The process is shown in Figure 4.

Spurious signals are divided into static and dynamic spurious waves. Static clutter refers to radar signals in the environment that are reflected by static targets. Dynamic interference comes from echoes from other moving parts of the hand. From the perspective of parametric characteristics, static target echoes are quite different from moving target echoes, which tend to exhibit low-frequency characteristics in the Doppler domain and thus can usually be achieved by using high-pass filtering [49], pulse-to-cancellation [50], background subtraction [51], or adaptive clutter suppression algorithms [52] for static clutter filtering. For dynamic interference, traditional clutter suppression methods are mainly based on various types of constant false alarm algorithms (CFAR), which detect the clutter energy to determine the judgment threshold and then achieve the elimination of dynamic interference signals [53]. However, in recent years, some studies have found that the energy and amplitude of some dynamic clutter signals are very similar to real gesture signals, which is called target-like clutter. They have serious uncertainty and time-varying characteristics, and the traditional CFAR method cannot solve the complex clutter interference. To address the above problem, some studies proposed solving the mixing problem of clutter signals and target gesture signals through location information. Xia et al. [54] proposed a preprocessing method for spatial position alignment to improve the spatial consistency of a multi-position dataset. There are also studies that target specific forms of clutter interference by mathematically modeling them to achieve the cancellation of target-like clutter. Ritchie et al. [55] used deep learning techniques to exploit the joint location-energy feature information of real gesture targets and target-like targets on the distance-angle spectrum to achieve clutter suppression.

### 4.2. Feature Extraction

In the previous section, we introduced RDM, which is also one of the most commonly used gesture features. However, the RDM only contains the position and velocity of the target in a single-frame signal and cannot represent the complete gesture feature over a period of time. Therefore, some features with a time dimension are used to represent the overall gesture. In fact, classical gesture features can be divided into time-frequency maps and spectrum map videos. Time-frequency maps include range-time maps (RTMs), Doppler-time maps (DTMs), and angle-time maps (ATMs). The spectrum map videos consist of the multi-frame accumulation of range-Doppler maps (RDMs), range-angle maps (RAMs), and Doppler-angle maps (DAMs) [29]. These features reflect information about the target in terms of distance, Doppler, and angle. The time-frequency maps contain the trajectory of a feature over time. We introduced how to obtain the range and Doppler information in the previous chapter, and RTM and DTM can be constructed by accumulating the Range-bin and Doppler-bin. For example, using the distance information obtained by weighted averaging each 2D range map and then stitching each frame in turn, the RTM of this gesture signal can be obtained. Similarly, the velocity vector at the distance unit of the target on each RDM frame is extracted, and the velocity vectors of each frame are stitched together to obtain the DTM of the gesture signal. Different from distance and Doppler information, angle information is calculated from the data acquired by the different receiving antennas. There are many different algorithms for angle estimation, such as Beam-Forming, Minimum Norm, Multiple Signal Classification (MUSIC) [56], etc. Beamforming is a signal processing technique used to estimate the angle of arrival (AoA) of incoming signals by combining the signals received from multiple antennas in a specific way. The Minimum Norm method estimates the direction of arrival of a signal by minimizing the power of the received signal, subject to some constraints, and provides superior interference rejection compared to conventional beamforming. MUSIC is a high-resolution algorithm for direction finding and source location in array signal processing. It estimates AOA by analyzing the eigenvalues and eigenvectors of the received signal covariance matrix. MUSIC not only provides excellent angle resolution, but also has good robustness to noise and has been widely used in gesture recognition. Yao et al. [57] used the MUSIC algorithm to extract the angle feature and construct the angle-time map (ATM) of multi-hand gestures. The spectrum map videos contain more feature information in each frame of the map. Similar to RDM, RAM and DAM are maps calculated from one frame of radar data in two different dimensions. Currently, features that can represent distance, velocity, and angle information, such as RDM [58,59], RAM [12], DAM [28], RTM [60,61], DTM [62], and ATM [63], have been widely used in the research of gesture recognition. 

However, it is difficult to achieve high recognition accuracy by relying on a single feature in gesture recognition. In contrast, rich gesture features can improve the accuracy of a recognition system. Therefore, many studies have proposed feature fusion as a way to consider more gesture features in gesture recognition. Since the traditional convolutional neural network structure is limited by a single input data set, some studies have proposed a multi-channel algorithmic model to exploit more gesture features [64,65]. In addition to multi-channel networks, some studies have proposed feature stitching methods to create new multidimensional features, such as constructing multi-feature cubes including range, Doppler, and angle features, which are input classification models with one channel. We selected several representative studies that have applied feature fusion methods, as shown in Table 2. We counted the type and number of gestures, the classification algorithm, and the accuracy of using different features in these studies. Due to the differences in the gestures, datasets, and classification algorithms used in these studies, it is meaningless to directly compare the results of different studies, but there are some commonalities between these studies under different conditions. By comparing the results of these studies applying different characteristics in their experimental settings, we summarized the findings and recommendations as follows:

Before comparing the results of different feature extraction methods, it is necessary to clarify the radars used in these studies and their resolution in different dimensions. In gesture recognition, resolution has the greatest correlation with features. As we mentioned before, the object of gesture recognition is the movement of dynamic gestures that are reflected by features. Higher resolution may allow more gesture motion details to be included in the same feature, which is more important in the detection of micro-gestures. Based on radar parameters, we calculated the range, velocity, and angular resolution of the radars used in the studies according to the formula, as shown in Table 2. It can be noticed that the studies generally set the resolutions at a high level, and because the configurations of the radars used are very similar, the values of the resolutions are also very close to each other. In the recognition of most gestures, including micro-gestures, these small differences cannot produce significant differences or effects in features. In addition, the detection accuracy is different from the resolution, which can be affected by the frequency resolution of the processing algorithm. It is often possible to increase the frequency resolution of the algorithm by increasing the length of the sampling sequence, but this also reduces the efficiency of the algorithm. In fact, it is usually not necessary to detect the specific position, speed, or angle value in gesture recognition; rich and detailed features are the most important.

Excluding the effect of resolutions on the results, we are able to directly compare each feature extraction method. First, by comparing the results of applying a single feature, it can be found that the accuracy of the feature in the recognition of different gestures varies greatly, even in the same study. We believe this is because the expressive force of gestures in different dimensions is different. For example, the change of range and Doppler in macro gestures is more prominent, while angle information plays a greater role in the recognition of micro gestures. This confirms, from another perspective, the need to consider multiple characteristics. In addition, by comparing the results of applying a single feature and a fusion feature, it can be found that the application of a fusion feature can increase the accuracy of gesture recognition regardless of the experimental conditions, and the accuracy tends to increase with the increase in feature information. It is worth noting that we also found that there were cases where the feature information increased but the accuracy did not increase or even slightly decreased [68], although these cases were very rare. We believe that this is because the integrated features with less feature information have fully expressed the gestures to be measured, and the continued addition of feature information may cause redundancy, which affects accuracy to a certain extent. Finally, in the statistical analysis of the methods and results of the two types of feature fusion, we found that methods that put multiple features on the same map or feature cube tend to have slightly higher accuracy compared to multi-channel feature fusion methods, and similar results were found in the study [40]. We believe this is due to the fact that the former reduces complexity while enriching gesture features and, at the same time, enhances the correlation between features of different dimensions. With the above discussion and results, we suggest that researchers should consider features of multiple dimensions based on the method of feature fusion in feature extraction, including but not limited to range, Doppler, and angle. Of course, this should also consider other experimental and application conditions.

### 4.3. Datasets

In this section, we conducted statistics on the datasets in gesture recognition. Since most studies used self-constructed data sets, we selected some representative self-constructed datasets for summary and analysis, as shown in Table 3. Currently, there are limited publicly available resources for comprehensive radar gesture datasets, except for the freely accessible Google soli dataset [25], and only a few studies have collected and made public comprehensive gesture datasets with large samples and data volumes, such as the Dop-NET [69] and M-Gesture [70] datasets. The Soli dataset comes from Google’s Project Soli sensor. The dataset contains a total of 5500 gesture samples, and gesture movement is represented by four RDMs. These data are divided into two parts: One part contains a sample of 11 gestures from 10 experimenters, with 25 samples of each gesture and a total sample size of 2750. The other part contains 2750 samples generated by a single experimenter in six gestures, which can be used in some comparison experiments. Dop-Net is a large radar database organized in a hierarchy in which each node represents the data of a person. This data was obtained using FMCW and CW radars. Unlike the Soli dataset, the radar signal in Dop-NET is based on micro-Doppler signatures. The dataset includes data from four different gestures (Wave/Pinch/Click/Swipe) from six experimenters, for a total sample size of 3052. M-Gesture is a large gesture dataset built by Liu et al. [70]. The data was obtained from 144 experimenters, for a total sample size of 56,420. The dataset, obtained from Radar IWR1443, contains a total of 14 gestures and includes a variety of data types such as eigenvalue sequences, RDM, point clouds, and raw data. In addition, the data was divided into two experimental scenarios: short-range and long-range, and the gestures, number of samples, and experimenters differ in both scenarios, allowing the dataset to meet the needs of a wider range of experimental conditions.

The self-constructed datasets used in most of the studies differed considerably in terms of features, the number of experimenters, the type of gestures, and the total sample, as shown in Table 3. Through statistics, it can be found that the number of experimenters and gesture types in the self-constructed datasets varies from a few to several dozen. These datasets are subject to different experimental and application conditions, but there is no doubt that datasets with more experimenters and gesture types tend to have better generalization and robustness of the algorithms, although they face greater recognition difficulties. Most of the samples of a single gesture in the self-constructed datasets are in the range of dozens, which shows that the number of samples per gesture in this magnitude is adequate for the experimental needs. It is also worth noting that in some studies, not all of the datasets were scaled for training and testing, but rather a small number were tested independently as ‘unfamiliar users’ or ‘unfamiliar gestures’, and some studies may even use all of the ‘unfamiliar user’ data for testing. This allows for a good evaluation of the robustness and generalizability of the algorithm, which are of great importance in practical applications.

### 4.4. Classification Algorithms

In this section, we summarize the classification algorithms in the studies on gesture recognition and count the experimental conditions (number of gestures and experimenters, total sample, features) and results where the classification algorithms were located, as shown in Table 4. Similar to the analysis of feature fusion methods in Section 4.2, due to the different experimental conditions of classification algorithms, we cannot judge the advantages and disadvantages of different classification methods by directly comparing the results of different studies, but we can find the common values or rules from the statistics.

The widely used machine learning methods in gesture recognition are support vector machine (SVM) [27,73], K-nearest neighbor method (KNN) [66], and hidden Markov model (HMM) [74]. These methods are easy to implement but lack robustness and computational efficiency. Especially when the gestures are complex and the training sample size is large, the accuracy rate will drop significantly. Some research has given suggestions and methods to improve the above algorithm [79,80]. The Dynamic Time Warping (DTW) algorithm can deal with the similar relationship between the time series of two gestures well. However, the DTW algorithm also has the limitations of high computational complexity and poor robustness [44,75]. Additional studies have enhanced the DTW algorithm for gesture recognition by adding path constraints and refining the matching procedure for gesture recognition [81]. According to the data in statistics [27,44,66,73,75], it can be found that machine learning algorithms are more suitable for problems with simple gestures, small sample categories, and quantity in gesture recognition, and deep learning algorithms tend to have better performance in gesture recognition problems with complex gestures and a larger number of samples.

Classification algorithms based on deep learning have been widely used in gesture recognition, which mainly involves two classical deep neural network models: the convolutional neural network (CNN) [49] and the long short-term memory network (LSTM) [50]. From the data in the statistics, it can be found that CNN, 3D-CNN, and CNN-LSTM are often used in gesture recognition. 3D-CNN is an extension of traditional CNN in the time dimension that can capture both spatial and temporal features in 3D data. In gesture recognition, 3D-CNN often has better performance than CNN, especially when the features are spectrum map videos (RDM, RAM, and DAM). However, it also has a higher computational cost compared to CNN. In addition, there are many studies that have applied CNN-LSTM networks [31,76,77], which benefit from the strengths of both CNN and LSTM. It uses CNN to learn spatial features from input data and then feeds these features to LSTM to learn temporal patterns. It usually has higher recognition accuracy than using LSTM or CNN algorithms alone, making it suitable for recognizing complex gestures. Neural networks with multiple channels are a common optimization method that increases the feature information input to the classification model, which has been discussed in Section 4.2. In addition, neural networks with optimized architectures such as I3D, S3D, VGG-Net, Res-Net, Dense-Net, and Transformer have also been used in gesture recognition, often with higher accuracy. These methods achieve better performance by optimizing the network in terms of convolutional kernel, depth, width, connection mode, and mechanism. I3D extends the traditional CNN from 2D to 3D, similar to 3D-CNN, but it is also a dual-stream network. S3D introduces efficient 3D convolutions using separable spatial and temporal convolutions, reducing computational complexity while maintaining spatio-temporal modeling. VGG-Net demonstrated the importance of using deeper networks and smaller convolutional filters to learn more complex features. The key improvement in ResNet is the introduction of skip connections (residual blocks), enabling the training of extremely deep networks with hundreds or even thousands of layers. Dense-Net introduces the concept of dense connectivity, connecting each layer to every other layer in a feed-forward manner, promoting feature reuse and information flow. Transformer’s key idea is the self-attention mechanism, which allows the model to capture dependencies between different positions in the input sequence effectively. In general, from spatial-temporal modeling to efficient convolutions, from deeper architectures to residual connections, and from dense connectivity to attention mechanisms, these ideas have significantly advanced the capabilities and efficiency of neural networks in gesture recognition. In addition, some studies have incorporated attention block into neural networks, an approach that further reduces the effects of noise and clutter and adaptively focuses on important features and suppresses unnecessary ones [29,66]. There are also studies here that incorporate deformable blocks [29] or spatiotemporal deformable convolution (STDC) blocks [40] into neural networks. This method is able to improve the motion modeling ability of gestures by learning extra offsets, improving the generalization of the algorithm and the accuracy of recognizing complex gestures. In addition, Zhao et al. [40] proposed an adaptive spatiotemporal context-aware convolution (ASTCAC) block to improve the ability of the recognition network to capture both global and local contextual information. These optimization algorithms for classification models all contribute to the improvement of accuracy in gesture recognition, which is also confirmed by the statistical results in Table 4. In general, deep learning-based classification algorithms can provide high accuracy and robustness. However, these methods also tend to face problems such as high complexity and high computational costs. Based on the above discussion and results, we suggest that researchers select appropriate classification algorithms according to the experimental conditions. For example, in the case of simple gestures and a small sample size, machine learning or 2D-CNN algorithms can be chosen with spectral maps as features. In the opposite case, researchers can choose a multi-channel neural network with fusion features and use some optimization architectures or methods, such as attention or deformable blocks.

### 4.5. Generalization

Generalization is the focus of improving accuracy in gesture recognition, which can make the recognition system perform better in the face of strange people and complex environments. In this part, we summarized the suggestions and methods to improve algorithm generalization in gesture recognition and discussed three aspects.

Firstly, abundant samples and sufficient data are important factors in improving the accuracy of gesture recognition. There are some studies that have proposed data augmentation. The purpose is to increase the number of samples and improve the recognition effectiveness of the model. Data augmentation can be broadly classified into two types: generative adversarial networks (GANs) and mixup augmentation (MA). The images generated by the GAN are not derived from the original samples but are mainly trained by the model to obtain the applicable images, thus increasing the number of samples [82]. The MA algorithm is able to increase the number of features by random cropping, translation transformation, scale transformation, contrast transformation, and rotation transformation based on the original image [66]. It is also able to fuse features from multiple types of samples to increase the diversity of the samples. In general, the MA avoids overfitting while improving the generalization ability of the model and thus improving the gesture recognition rate [67].

Secondly, we focused on data dependencies in gesture recognition. Indeed, most gesture recognition methods require a large amount of data for training. Studies generally focus on the classification of the fixed predefined gestures that already exist in the training set, but gesture data in real scenes often contains more variations. In recent years, several studies have proposed the use of meta-learning networks as a solution to the few-shot learning problem for the sample and generalization problems of gesture recognition. Different from conventional machine learning or deep learning models, the proposed model is able to make use of domain knowledge learned from a relatively large number of labeled points to quickly adapt to unseen hand gesture classes with only a few training observations [29,83]. This method not only adapts well to the new environment but also solves the data dependency problem and reduces computational complexity. In addition, unsupervised network-based gesture recognition methods have also received attention [35]. These methods automatically classify dynamic gesture datasets without labels, using the intrinsic closeness of the data. They are more generalizable and efficient than classification methods relying on labeled data. However, it is difficult to apply radar data.

Thirdly, the variability of gestures is also an important direction to take to solve the generalization problem. In practical applications, the recognition of new users is an unavoidable problem for gesture recognition systems. Users with different hand habits and diseases (e.g., Parkinson’s disease) often pose a great challenge to gesture recognition systems. Zeng et al. [84] applied the above meta-learning network-based approach to dynamic gesture recognition, which solved the problem of new user-defined gestures while achieving good performance. In addition, the characteristic trajectory and spatial location of the gesture are important factors in discerning gesture variability [31]. Xia et al. [54] proposed a spatial position alignment method to improve the spatial consistency of a multi-position dataset and the generalization performance of gesture recognition by using multi-dimensional position spectrum features. These methods are also commonly used in the recognition of handwritten trajectories and patterns [71].

## 5. Challenges

### 5.1. Gesture Recognition in Complex Environments

The experimental scenario of the existing study is ideal without much interference compared to the actual complex application environment. However, in long-distance or large field-of-view (LD/LFoV) environments, there are not only more interference factors but also problems such as dynamic blurring of target gestures due to small observation angles [68]. Although studies have focused on this aspect [54], there is still a great research prospect. 

On the other hand, most studies have used single radar sensors for dynamic gesture recognition. However, in more complex application scenarios, the gesture information obtained by a single radar sensor is not rich and accurate enough. At present, studies have been conducted on incoherent radar sensor networks and joint recognition of radar and other categories of sensors [58,85]. The results showed that the application of multiple sensors can ensure a more accurate and stable gesture recognition system, but how to remove mutual interference between sensors and how to perform effective data fusion remains a challenge in this field.

### 5.2. Real Time and Complexity of Gestures

At present, gesture recognition tends to achieve high accuracy through a large amount of feature data and complex classification models, which require a large amount of memory and computing resources. However, most commercial embedded systems, such as smartphones and other portable devices, have limited memory and computing power. Complex gesture recognition systems are also unable to meet the requirements of real-time performance, which is not accepted in commercial applications. Therefore, many studies hope to reduce the complexity of gesture recognition systems while ensuring accuracy and put forward some methods from two aspects: data and classification models. The feature cube introduced in Section 4.2 is one of the main methods to reduce data complexity and improve data processing efficiency. In addition, the sparse signal processing technique provides a new way to reduce data complexity without affecting performance. Li et al. [42] proposed a sparsity-driven method for dynamic gesture recognition, which is expected to achieve real-time processing in practical applications. Due to the high complexity of the conventional deep neural network, a lot of computational energy is needed in gesture recognition, and the majority of the energy is consumed by the multiply-accumulate (MAC) operations between layers. Therefore, researchers have proposed to reduce the computational complexity by using lightweight networks [86,87] or by optimizing the classification model through methods such as pruning techniques [58,67]. In general, reducing the complexity of the algorithm while ensuring accuracy and meeting the real-time requirements of the application is a great challenge for gesture recognition and will also be the focus of future studies.

## 6. Conclusions

The progress of FMCW mmW radar gesture recognition technology opens up a new approach to human-computer interaction. This paper summarized the methods, results, and key issues of gesture recognition based on the main step process of gesture recognition, discussed three aspects of feature fusion, classification algorithms, and generalization, and provided analysis and recommendations for other researchers based on the statistical data. This paper provides a reference for future research on gesture recognition based on FMCW mmW radar and is of great significance in promoting the practice and research of gesture recognition methods.

## Figures and Tables

**Figure 1 sensors-23-07478-f001:**
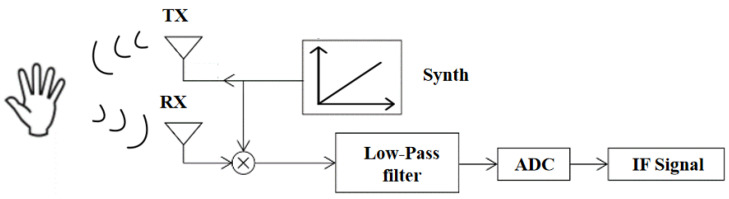
Date acquisition processing.

**Figure 2 sensors-23-07478-f002:**
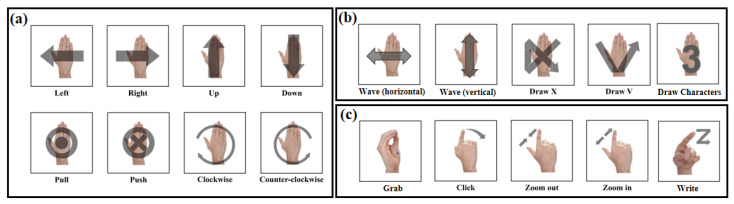
Gesture summary and classification: (**a**) macro gestures with single-direction, (**b**) macro gestures with multi-direction, and (**c**) micro gestures.

**Figure 3 sensors-23-07478-f003:**
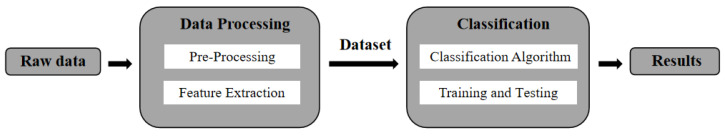
Process diagram of data processing and classification.

**Figure 4 sensors-23-07478-f004:**
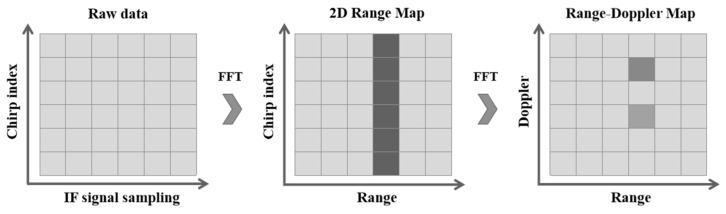
The process of obtaining RDM.

**Table 1 sensors-23-07478-t001:** FMCW mmW radar usage statistics.

Model	Frequency Coverage	Available Bandwidth	Number of Antennas	References
I/AWR1642BOOST	76~81 GHz	4 GHz	2TX, 4RX	[12,27,28]
I/AWR1843BOOST	76~81 GHz	4 GHz	3TX, 4RX	[29,30]
I/AWR1443BOOST	76~81 GHz	4 GHz	3TX, 4RX	[31,32]
I/AWR6843	60~64 GHz	4 GHz	3TX, 4RX	[33,34]
BGT60TR13C	58~63.5 GHz	5.5 GHz	1TX, 3RX	[35,36,37]
BGT24MTR12	24~24.2 GHz	200 MHz	1TX, 2RX	[38]

**Table 2 sensors-23-07478-t002:** Statistics of methods and results in feature extraction.

Radar	Resolution			Gestures/Number	Classification Algorithm	Single Features/Average Accuracy	Fusion Features/Average Accuracy		Reference
	Range	Velocity	Angular				One-Channel	Multi-Channel	
AWR1642 BOOST	3.95 cm	0.2 m/s	≥0.5°	Macro-gestures/7 Micro-gestures/3	SVM	RTM/70.69% DTM/87.59% ATM/69.28%	RTM + DTM/89.43% DTM + ATM/98.15% RTM + ATM/91.37% RTM + DTM + ATM/98.48%	\	[27]
IWR1642 BOOST	3.75 cm	0.032 m/s	≥0.5°	Macro-gestures/7 Micro-gestures/3	CNN	DTM/91.34%	3D-festure/96.61%	\	[66]
IWR1642 BOOST	8.33 cm	0.19 m/s	≥0.5°	Macro-gestures/9 Micro-gestures/1	3D-CNN	RTM/91.60% DTM/92.80% ATM/92.10% RDM/93.20% DAM/93.90%	Feature Cube (RDAM) 98.10%	\	[28]
IWR1443 BOOST	3.75 cm	0.8 m/s	≥0.5°	Macro-gestures/6	CNN	RTM/89.6% DTM/87.3% ATM/84.3%	\	RTM + DTM + ATM/91.6%	[31]
AWR1642 BOOST	3.75 cm	0.4 m/s	≥0.5°	Macro-gestures/6	VGG-16	RTM/89.3% DTM/86.3% ATM/87.0%	\	RTM + DTM + ATM/92.0%	[67]
AWR1642 BOOST	3.75 cm	0.4 m/s	≥0.5°	Macro-gestures/6	DTW	RTM/89.50% DTM/89.83% ATM/88.50%	\	RTM + DTM + ATM/94.50%	[60]
AWR1642 BOOST	4.46 cm	0.4 m/s	≥0.5°	Macro-gestures/8	3D-CNN	RDM/72.16% RAM/82.79%	\	RDM + RAM/86.95%	[12]
AWR1843 BOOST	≥3.75 cm	\	≥0.5°	Macro-gestures/7 Micro-gestures/3	2D-ResNet18	RTM/83.70% DTM/88.63% ATM/60.37%	\	RTM + DTM/91.52% RTM + ATM/73.00% DTM + ATM/84.00% RTM + DTM + ATM/90.48%	[29]
AWR1843 BOOST	≥3.75 cm	\	≥0.5°	Macro-gestures/7 Micro-gestures/3	3D-ResNet18	RDM/92.26% RAM/87.07% DAM/91.37%	\	RDM + RAM/90.33% RDM + DAM/92.52% DAM + RAM/89.70% RDM + RAM + DAM/93.30%	[29]
AWR1843 BOOST	≥3.75 cm	\	≥0.5°	Macro-gestures/7 Micro-gestures/3	2D + 3D ResNet18 (Dual-flow)	\	\	DTM + RDM/93.70% DTM + DAM + RAM/94.96% DTM + RDM + DAM + RAM /95.63% RTM + DTM + RDM/94.11% RTM + DTM + DAM + RAM /95.22% RTM + DTM + RDM + DAM + RAM 96.04%	[29]
AWR1642 BOOST	3.75 cm	0.2 m/s	≥0.5°	Macro-gestures/16	2D/3D-CNN	\	5D feature cubes/99.53%	RTM + DTM/92.47% RTM + DTM + ATM + ETM/98.87%	[40]

**Table 3 sensors-23-07478-t003:** Statistics of the datasets.

Dataset	Radar	Features	Experimenters/Number	Gestures/Number	Samples of Each Gesture	Total Sample	Reference
Soli	BGT60TR13C	RDM	10	11	25	2750	[25]
Dop-Net	Ancortek radar	DTM	6	4	\	3052	[69]
M-Gesture	IWR1443 BOOST	Eigenvalue sequences, RDM, Point Cloud and Raw data	144 (64 men and 80 women)	14	10/15/30/50	56,420	[70]
Self-constructed	IWR1642 BOOST	RTM, DTM, ATM	5	10	30	1500	[28]
Self-constructed	AWR1642 BOOST	RDM, RAM	5	8	100	4000	[12]
Self-constructed	AWR1843 BOOST	RTM, DTM, ATM RDM, RAM, DAM	9	6	50	2700	[29]
Self-constructed	AWR1642 BOOST	Feature Cube	19	16	65	19,760	[40]
Self-constructed	BGT60TR13C	Feature Cube	20	12	30	7200	[71]
Self-constructed	AWR1642 BOOST	RDM, RTM, DTM, ATM	8 + 2	7 + 1	50	4000	[68]
Self-constructed	BGT60TR13C	RDM	9 + 9 + 10	20 + 15 + 14	\	3696 + 2788 + 1934	[72]

**Table 4 sensors-23-07478-t004:** Statistics of classification algorithms and results.

Gestures/Number	Experimenters/Number	Total Sample	Features	Classification Algorithms	Accuracy	Reference
7	5	1750	DTM and the phase spectrum	SVM	93.84%	[73]
6	2	1250	RTM + DTM + ATM	SVM	98.48%	[27]
4	2	1200	DTM	HMM	83.3%	[74]
10	5	1050	RTM	DTW	91%	[44]
12	10	1200	DTM	DTW	93.5%	[75]
10	10	5000	DTM	KNN SVM CNN	88.93% 90.21% 91.34%	[66]
10	10	5000	3D Feature	CNN CNN + Attention Module	96.61% 97.17%	[66]
7	10	4200	RDM	RNN CNN 3D-CNN	90.27% 93.58% 99.06%	[51]
8	5	4000	RAM RDM + RAM	3D-CNN 3D-CNN (Multi-Channel)	82.79% 86.95%	[12]
8	10	1600	RDM, RAM	CNN-LSTM	94.75%	[76]
6	4	2400	RDM	3D-CNN CNN-LSTM	95% 97%	[77]
5	9	4500	RTM, DTM, ATM	LSTM CNN-LSTM	96.7% 99.6%	[31]
10	\	4000	RDM	CNN 3D-CNN LSTM I3D I3D + LSTM	82.77% 88.07% 90.35% 89.37% 93.05%	[78]
49	28	8418	RDM	Transformer	93.95%	[72]
6	9	2700	RTM, DTM, ATM	VGG-19 ResNeXt101 DenseNet161	93.52% 93.33% 92.69%	[29]
6	9	2700	RDM, RAM, DAM	S3D I3D 3-D ResNeXt152	95.37% 94.54% 95.19%	[29]
6	9	2700	RTM, DTM, ATM RDM, RAM, DAM	2D/3D-ResNet18 + Deformable + Attention	97.52%	[29]
16	19	19,760	5D Feature cube	S3D S3D + STDC S3D + ASTCAC S3D + STDC + ASTCAC	98.80% 99.12% 99.01% 99.53%	[40]

## Data Availability

No new data were created.

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
