# Peer review of "Dynamic Gesture Recognition Based on FMCW Millimeter Wave Radar: Review of Methodologies and Results"

_sensors, 2023, doi:10.3390/s23177478_

Round 1

Reviewer 1 Report (Previous Reviewer 2)

In the reviewed version of the manuscript, the author has addressed the most critical concerns of the reviewer. Now the paper is technically fine. Therefore, it seems that the present form of this paper has the potential for publication in Sensors Journal.

Sincerely, 

The reviewer

 Date of this review

July 6, 2023 

Moderate editing of English language required

Author Response

Reviewer 2 Report (Previous Reviewer 1)

The paper looks more like an essay with the names of different types of algorithms with no in-depth scientific analysis identifying the relative merits and demerits of different gesture recognizing algorithms. The major shortcoming of the paper is the complete absence of the accuracy and resolution of range, velocity, angle (azimuth and elevation), and object separation capability of the FMCW radars in question. Transmit power, receive power, and minimum detectable range of the relevant radars are absent as well. The authors compared the accuracy of different algorithms in Tables 2-4 without specifying what type of radars were used and what were the specifications and accuracies of those radars.  Without considering the accuracy and resolutions of the test radars, it is meaningless to compare the accuracy of data from various algorithms as the algorithm accuracy is not decoupled from the radar hardware accuracy. The authors presented Different methods and algorithms, however, the scientific differences among them were not reviewed. Resolution is another important factors, which is missing as well. Thanks to the authors for their effort, but it’s a long way to go. I recommend the paper to be rejected.

Author Response

Reviewer 3 Report (New Reviewer)

This paper titled "Dynamic Gesture Recognition Based on FMCW Millimetre Wave Radar: Review of Methodologies and Results" by Gaopeng Tang et al. presented a review of radar-based gesture recognition. The manuscript is overall interesting and may be helpful for the gesture recognition community. However, the authors need to revise the manuscript with some respects.

1.     The principle of ranging is not explained. The authors need to describe the equations of the distance, the ranging resolution, the ranging accuracy etc. Generally, the ranging accuracy is expressed with Cremer Rao lower bound (CRLB). Derivation of the CRLB inequation is also preferable.

2.     The authors mentioned “laser radar” in page 2, but there was no explanation about it. I suggest the authors to compare gesture recognition using laser radar (LiDAR, depth camera) such as the Kinect, RealSense, LiDAR on iPhone/iPad, with that using radar.

3.     Besides, the refresh rate and the horizontal/vertical resolution/accuracy of radar for gesture recognition are not described.

4.     How do this type of radar measure 3D shapes? Is it done by beam steering using phased array? In Figure 1, the beam scanning part.

5.     Acquisition of angle information is done by calculation of data from different receivers.

Explain more about beam-forming, minimum norm, MUSIC etc.

Minor editing of English language required

Author Response

Reviewer 4 Report (New Reviewer)

This paper summarized the many methods and challenges involved in recent research on Frequency Modulated Continuous Wave (FMCW) radar-based gesture and pattern recognition. This paper is of interest to the radar research community where there are several issues need to be resolved on gesture recognition technology.

However, a few minor issues need to be resolved before this paper can be published.

1)       On page 1, line 6 from the top, please define the full form of FMCW before using the abbreviation.

2)       There are several grammatical errors. For an example, on page 2, line 24, the description reads as, “In this review we searched several databases for references, covered following databases…”, where it should read as, “In this review article, we searched several databases for references, and covered following databases:,

Minor edits. 

Author Response

This manuscript is a resubmission of an earlier submission. The following is a list of the peer review reports and author responses from that submission.

Round 1

Reviewer 1 Report

The paper is more like a general summary of FMCW radar-based various gesture recognition methods without any scientific merit. No quantified data-based  comparison of the methods is provided. The operation of an FMCW radar is described without specifications.

Reviewer 2 Report

Major Comments and Suggestions for Authors of Sensors-2384389 manuscript.

 The manuscript summarized the relevant literature on gesture recognition using FMCW millimeter-wave radar from January 2015 to January 2023.

-Reference datasets in the field of gesture recognition using FMCW millimeter-wave radar have not been reviewed in this manuscript.

- The main goal of the review manuscripts is that the authors can provide the reader with the best solution for specific sensors with different conditions and limitations. Such a summary is not seen in this manuscript. Also, the conclusions presented in the manuscript should be based on conducting experiments on different datasets and providing quantitative and qualitative results from the implementation of the proposed methods on these referenced datasets, which is not addressed in this manuscript.

Therefore, this reviewer thinks that this paper should not be published in a Sensors journal before all of the above major corrections are applied by the authors in this manuscript.

Sincerely,

The reviewer

Date of this review

9 May 2023

Moderate editing of the English language is required.